# Activating Transcription Factor 3 (ATF3) Regulates Cellular Senescence and Osteoclastogenesis via STAT3/ERK and p65/AP-1 Pathways in Human Periodontal Ligament Cells

**DOI:** 10.3390/ijms26104959

**Published:** 2025-05-21

**Authors:** Won-Jung Bae, Sang-Im Lee

**Affiliations:** 1Department of Pharmacology, College of Dentistry, Dankook University, Cheonan 31116, Republic of Korea; 2Department of Dental Hygiene, College of Health Science, Dankook University, Cheonan 31116, Republic of Korea

**Keywords:** ATF3, cellular senescence, human periodontal ligament cell, osteoclast differentiation, periodontal diseases

## Abstract

Oral cellular aging plays a critical role in the pathogenesis of chronic periodontitis and alveolar bone resorption. Although activating transcription factor 3 (ATF3) has been implicated as a senescence-associated factor, its specific role in periodontal ligament cell (PDLC) senescence remains unclear. Human PDLCs were exposed to lipopolysaccharide (LPS, 1 μg/mL) and nicotine (5 mM) for 3 days to induce senescence. ATF3 expression was silenced using siRNA. The expression of senescence-associated secretory phenotype (SASP) factors (IFNγ, IL6, IL8, TNFα, and IL1β) and the secretion of nitric oxide (NO) and prostaglandin E_2_ (PGE_2_) were assessed by RT-PCR and immunoassay. Conditioned media (CM) from these cells were applied to mouse bone marrow macrophages (BMMs) to evaluate osteoclast differentiation and bone resorption. In addition, key signaling pathways, including STAT3, ERK, NF-κB (p65), and AP-1, were investigated by Western blotting and immunofluorescence. ATF3 knockdown markedly reduced the LPS/nicotine-induced expression of SASP factors and decreased NO and PGE_2_ levels. CM from ATF3-silenced PDLCs markedly inhibited osteoclast differentiation, as evidenced by reduced tartrate-resistant acid phosphatase (TRAP)-positive multinucleated cells and diminished bone resorption. Moreover, ATF3 inhibition led to a decreased RANKL/OPG ratio and attenuated the phosphorylation of STAT3 and ERK, along with the reduced nuclear translocation of p65 and AP-1 components. These findings suggest that ATF3 plays a critical role in mediating cellular senescence and osteoclastogenesis in PDLCs. Targeting ATF3 may represent a novel therapeutic strategy for managing age-related oral diseases, such as chronic periodontitis.

## 1. Introduction

Aging is a progressive degenerative process influenced by intrinsic and extrinsic factors, including DNA damage, oxidative stress, and inflammation. Cellular senescence, a key mechanism of aging, contributes to functional decline and increased susceptibility to diseases [1,2]. In the oral cavity, a senescent microenvironment drives physiological and structural changes, accelerating periodontal disease and alveolar bone loss. Chronic inflammation, recognized as a hallmark of aging, has a cyclical relationship with age-related diseases, further exacerbating tissue degradation [3]. In addition to host factors, bacterial colonization and biofilm formation in the oral cavity are key contributors to chronic inflammation and tissue damage in periodontitis, highlighting the critical role of oral microbiota in disease progression [4]. Environmental stressors, like lipopolysaccharide (LPS) and nicotine, significantly contribute to oral cellular senescence by inducing oxidative stress and inflammation [5,6].

LPS, a component of Gram-negative bacterial membranes, triggers an immune response that leads to inflammation and reactive oxygen species (ROS) production [7]. This inflammatory milieu promotes cellular senescence and alveolar bone loss, facilitating the progression of periodontitis [8]. Nicotine, a key component of cigarette smoke, further exacerbates oxidative stress and inflammation, accelerating aging-related damage to oral and periodontal tissues [9]. Additionally, nicotine directly induces cellular senescence in gingival cells [10], worsening periodontal degradation [11].

Activating transcription factor 3 (ATF3), a member of the ATF/cAMP response element-binding protein (CREB) family, plays a crucial role in the cellular stress response. It acts as both a transcriptional activator and repressor, regulating genes involved in cell survival, apoptosis, and proliferation [12]. Recent studies highlight ATF3’s role in cellular senescence through chromatin reorganization [13], particularly in cardiac myocytes [14] and macrophages [15]. However, its involvement in oral cell senescence remains unclear.

This study aims to investigate the role of ATF3 in a cellular senescence environment induced by LPS and nicotine. Specifically, we seek to elucidate the mechanisms by which ATF3 mediates cellular senescence and inflammation in response to these environmental stressors. Understanding ATF3’s role in this context may provide insights into the molecular pathways driving aging and help identify potential therapeutic targets to mitigate the adverse effects of environmental stress on oral tissues.

## 2. Results

### 2.1. ATF3 Expression in Periodontitis Patients and Senescent PDLCs

To investigate the relationship between periodontal disease and ATF3 expression, we analyzed data from the NCBI GEO database (Figure 1A). The results indicated that, although not statistically significant, ATF3 was expressed at higher levels in patients with periodontitis compared to healthy individuals (left; accession number GSE27993). Additionally, although ATF3 expression markedly increased in chronic periodontitis patients, it is difficult to draw definitive conclusions due to the limited sample size of only one case (right; accession number GSE7321). When senescence was induced in a dose-dependent manner with LPS and nicotine, a significant increase in ATF3 protein levels was observed at 5 mM nicotine (Figure 1B). Therefore, subsequent experiments were conducted using LPS 1 µg/mL and 5 mM nicotine. To inhibit ATF3 expression induced by LPS and nicotine, siRNA was used. Among the concentrations tested (10~30 pmol), the 30 pmol treatment group showed a reduction in ATF3 expression (Figure 1C). Therefore, 30 pmol was selected for subsequent experiments.

### 2.2. Silencing of ATF3 Reduces SASP Factors in Senescent PDLCs

To determine whether ATF3 regulates the expression of cellular senescence phenotypes, the expression of SASP factors was assessed following treatment with LPS and nicotine. After suppressing ATF3 with siRNA, the expression of IFNγ, IL6, and IL8 genes, which were increased by LPS and nicotine treatment, decreased compared to the control siRNA-treated experimental group (Figure 2A–C). The expression patterns of mRNA levels of TNFα and IL1β genes were also similar (Figure 2D). Next, the secretion levels of the non-protein soluble SASP factors, NO and PGE_2_, present in the conditioned media (CM) collected from periodontal ligament cells (PDLCs) were measured. It was observed that after silencing ATF3, the total amount of NO and PGE_2_ in the CM induced with senescence by LPS and nicotine was lower compared to the CM obtained after treatment with LPS and nicotine alone (Figure 2E,F).

### 2.3. CM from ATF3-Silenced PDLCs Inhibits Osteoclast Differentiation and Bone Resorption

Experiments were conducted to investigate whether substances secreted by senescence-induced PDLCs with silenced ATF3 regulate osteoclast differentiation and bone resorption in alveolar bone. In the experiment, osteoclast differentiation was induced in mouse bone marrow-derived macrophages (BMMs) using a mixture of osteoclastogenic medium and CM from senescent PDLCs. Senescence was triggered in PDLCs by LPS and nicotine, with ATF3 expression silenced in some cells via siRNA before senescence induction. CM from ATF3-silenced senescent PDLCs significantly suppressed osteoclast differentiation in BMMs, while CM from senescent PDLCs without ATF3 knockdown promoted it. These findings suggest that ATF3 plays a crucial role in mediating the pro-osteoclastogenic effects of senescent PDLCs, and its silencing can attenuate the paracrine signals that promote osteoclastogenesis (Figure 3A). Additionally, a significant reduction was observed in the number of TRAP-positive multinucleated cells (MNCs) with three or more nuclei (Figure 3B). After treating mouse BMMs with CM obtained from PDLCs to induce osteoclast differentiation, the expression of NFATc1, a key transcription factor for osteoclast differentiation, and the genes for Cathepsin K and MMP9, which are important for osteoclast differentiation and activity, were assessed. The expression levels of Cathepsin K and MMP9 were reduced in the experimental group treated with CM derived from ATF3-silenced PDLCs stimulated with LPS and nicotine; whereas, NFATc1 expression showed no significant difference between the groups (Figure 3C). Next, to determine whether the senescence induction in PDLCs with regulated ATF3 could control osteoclast formation and differentiation, the gene expression of RANKL and OPG in PDLCs was compared. The comparative analysis of RANKL and OPG gene expression revealed that the expression of RANKL was suppressed, and the expression of OPG was increased in senescence-induced PDLCs with silenced ATF3 compared to those induced to senescence without ATF3 silencing (Figure 3D).

To investigate the effect of CM from senescence-induced ATF3-silenced PDLCs on osteoclast bone resorption activity, a bone resorption assay was conducted (Figure 4A). Mouse BMM cells were seeded on dentin slides and treated with CM to induce osteoclastic activity. Resorption pits represent discrete lacunae formed by osteoclasts, as they degrade the mineralized dentin surface. The resorption area reflects the cumulative surface area affected by osteoclastic activity. In our study, bone resorption by osteoclasts, indicated by the largest resorption area (Figure 4B) and the deepest resorption depth (Figure 4C), was observed in CM collected from PDLCs induced to senescence with LPS and nicotine. Both pit depth and resorption area were significantly reduced upon treatment with CM from ATF3-silenced PDLCs, indicating decreased osteoclast-mediated bone resorption.

### 2.4. ATF3 Knockdown Inhibits STAT3 and ERK Phosphorylation and p65 and AP-1 Translocation in PDLCs

Experiments were conducted to identify the mechanisms associated with the regulation of SAPS, osteoclast differentiation, and activity in senescence-induced PDLCs with controlled ATF3 expression. After the inhibition of ATF3 expression, the phosphorylation of STAT3 activated by LPS and nicotine in PDLCs was reduced (Figure 5A). Using the immunofluorescence assay, it was observed that the translocation of phosphorylated STAT3 from the cytoplasm to the nucleus decreased in ATF3-silenced PDLCs treated with LPS and nicotine (Figure 5B). Similarly, the phosphorylation of ERK was also reduced (Figure 5A). To confirm the expression of other transcription factors associated with senescence and osteoclastogenesis, including p65 and AP-1, such as c-Fos and c-Jun, their nuclear localization in PDLCs was examined. The translocation of p65 (Figure 5C,D) and c-Fos (Figure 5C) from the cytoplasm to the nucleus increased in response to LPS and nicotine stimulation but decreased in ATF3-silenced PDLCs. In contrast, c-Jun translocation showed no significant difference between siScr and siATF3 (Figure 5C); however, immunofluorescence confirmed a decrease in c-Jun nuclear localization (Figure 5D). 

## 3. Discussion

Cellular senescence and aging are closely linked, with accumulated senescent cells reducing tissue regeneration and promoting chronic inflammation, accelerating aging-related diseases [1,16]. SASP factors, including inflammatory cytokines and non-protein soluble factors, contribute to this process. LPS and nicotine are known to induce senescence in various cells, including PDLCs [17], though their direct role in PDLC senescence remains unclear. Our study confirmed that LPS and nicotine exposure significantly increased SASP factors, such as IFNγ, IL6, IL8, TNFα, and IL1β, along with NO and PGE_2_ secretion, supporting their role in PDLC senescence.

LPS is well-known as an external infectious agent that can directly induce senescence in various cell types, including endothelial cells [18], osteocytes [19], and dental pulp stem cells [20,21]. Similarly, nicotine has been reported to induce senescence in various cells, such as epithelial cells [22,23,24] and gingival fibroblasts [7,25]. Our previous studies have confirmed the ability of LPS (1 μg/mL) and nicotine (5 mM) to induce inflammatory responses and cell senescence in periodontal and other oral cells. For example, 1 μg/mL LPS combined with various concentrations of nicotine has been widely used to mimic chronic inflammatory conditions in PDLCs [26]. Furthermore, treatment with low, subcytotoxic concentrations of stimulants for 72 h can induce stress-mediated premature senescence in normal human cells, such as fibroblasts, due to excessive cellular stress [27]. Despite substantial evidence, it remains unclear whether LPS and nicotine are direct causative agents of PDLC senescence. Therefore, we directly induced cellular senescence in PDLCs using LPS and nicotine. As a result, the expression of inflammatory SASP factors, including IFNγ, IL6, IL8, TNFα, and IL1β genes, as well as the levels of non-protein soluble factors, such as NO and PGE_2_, were all significantly increased. Recent studies have reported that ATF3 induces senescence in human umbilical vein endothelial cells (HUVECs) [13] and cardiac myocytes [17]. These findings suggest a broader role for ATF3 in cellular senescence across various cell types, which implies that ATF3 could also play a crucial role in the senescence of PDLCs in our study. When we induced cellular senescence in PDLCs using LPS and nicotine after inhibiting the expression of ATF3, the previously increased expression of SASP factors, such as TNFα, IL1β, and IL6 genes, along with the secretion levels of NO and PGE_2_, were all significantly reduced. These results suggest that inhibiting ATF3 can suppress the onset of senescence in PDLCs, as we had hypothesized.

Oral aging exacerbates inflammatory responses in periodontal tissues, leading to increased alveolar bone resorption and higher susceptibility to conditions like periodontitis. This chronic inflammation contributes further to bone loss around teeth, which can result in eventual loss if left untreated [28]. MMPs (matrix metalloproteinases) degrade collagen and extracellular matrix in periodontitis, inducing the destruction of gingival connective tissue and alveolar bone. In our previous study, we demonstrated that the combined exposure to nicotine and LPS induces cytotoxicity and upregulates the expression of MMPs in PDLCs, which in turn, promotes osteoclastogenesis. These MMPs, particularly MMP-8 and MMP-9, contribute to the degradation of extracellular matrix components and facilitate the release of pro-osteoclastogenic factors, thereby exacerbating alveolar bone resorption [27]. Recent studies have revealed that ATF3 regulates the proliferation of osteoclast precursors [29] and calcium signaling [30], thereby inducing osteoclast differentiation. Additionally, during aging, ATF3/TFR1 induces ferroptosis in osteocytes, leading to cortical bone loss [31]. These findings suggest that the SAPS factors induced during cellular senescence in PDLCs may regulate osteoclast differentiation and activity through ATF3. This indicates that ATF3 could be a crucial transcription factor in controlling alveolar bone resorption and remodeling. Consequently, we aimed to investigate the effects of ATF3 inhibition on osteoclast differentiation and bone resorption in PDLCs undergoing senescence induced by LPS and nicotine.

In the present study, we aimed to examine the indirect effects on osteoclast differentiation using CM derived from senescent PDL cells in which ATF3 function was suppressed. Osteoclast formation was attenuated in BMMs treated with CM obtained from ATF3-silenced, LPS + nicotine-induced senescent PDL cells. These findings suggest that the composition of osteoclast-regulating factors within the CM was altered. Furthermore, we observed that the expression of osteoclast-related markers, such as Cathepsin K and MMP9 was similar in the untreated group, LPS + nicotine treated group, and siScr CM treated group. Although there was no significant change in NFATc1 expression, it was specifically decreased in the siATF3 CM group. This may be attributed to the significant increase in OPG expression observed only in the ATF3-silenced PDLCs, which we believe was sufficient to antagonize RANKL-induced osteoclastogenesis under inflammatory conditions. Additionally, we found that ATF3 knockdown reduced the RANKL/OPG ratio, a critical regulator of bone metabolism. A lower RANKL/OPG ratio suppresses osteoclastogenesis and promotes bone formation [32,33], further highlighting role of ATF3 in alveolar bone remodeling. However, our results have the limitation that the expression of SASP factors and RANKL/OPG was analyzed only at the gene level.

Although this study analyzed SASP factors and RANKL/OPG expression only at the gene level, it is considered a meaningful study showing the effect of ATF3-inhibited CM on osteoclast differentiation activity reduction in senescence-induced PDLCs.

We also investigated the signaling pathways involved in ATF3 mediated senescence and osteoclastogenesis. Although the expression of NFATc1, a key transcription factor for osteoclast differentiation in mouse BMM, did not vary in CM from PDLCs, RANKL expression increased in PDLCs. NF-κB/p65 and AP-1 are key regulators of RANKL expression and inflammatory responses [34]. Western blot analysis showed that p65 and c-Jun levels were similar or decreased in the siATF3-treated experimental group, and no activity was observed in response to LPS and nicotine treatment. In addition, immunofluorescence analysis showed that the decrease in nuclear c-Jun signal following ATF3 expression suppression was not statistically significant. This suggests that ATF3 plays a central role in SASP-mediated osteoclastogenesis by modulating these pathways. Although numerous factors have been identified to regulate osteoclast differentiation and formation, previous studies have highlighted the pivotal role of ATF3 in this process. Jeong et al. [29] reported that ATF3 promotes the activation of TRAP and NFATc1 promoters by interacting with NFATc1 and c-Fos. In addition, Fukasawa K et al. [25] demonstrated that the osteoclast precursor-specific deletion of ATF3 significantly reduced bone resorption and bone loss upon RANKL stimulation compared to controls. These findings suggest that ATF3 facilitates the proliferation of osteoclast precursors by enhancing cyclin D1 gene expression through AP-1-mediated mechanisms. Future studies are warranted to determine whether RANKL expression and secretion by senescent PDLCs are modulated by ATF3, and to clarify the role of RANKL-mediated signaling in the differentiation of BMMs into osteoclasts in this senescence model.

STAT3 and ERK, which play important roles in both inflammatory responses and bone-related diseases, are activated through different pathways but may influence the progression of bone-related diseases through cooperative interactions [35]. Our results showed increased phosphorylation of STAT3 and ERK following senescence induction, which was significantly reduced upon ATF3 silencing in senescent cell models. STAT3 translocation, heightened by LPS and nicotine, was also diminished in ATF3-deficient PDLCs. These findings indicate that ATF3 regulates inflammation and bone resorption via STAT3 and ERK signaling, reinforcing its potential as a therapeutic target for oral aging-related diseases, such as chronic periodontitis.

## 4. Materials and Methods

### 4.1. Gene Expression Profiling

To compare ATF3 mRNA expression in normal and diseased periodontal tissues, publicly available gene expression datasets for human samples were retrieved from the National Center for Biotechnology Information (NCBI) Gene Expression Omnibus database (accession numbers GSE27993 and GSE7321; http://www.ncbi.nlm.nih.gov/geo/ (accessed on 17 March 2011).

### 4.2. Cell Culture and Preparation of Condition Medium (CM)

Human PDLCs were purchased from ScienCell (catalog #2630, Carlsbad, CA, USA). Cells were cultured in α-MEM supplemented with 10% fetal bovine serum, 100 U/mL penicillin, and 100 μg/mL streptomycin in a humidified atmosphere of 5% CO₂ at 37 °C. Senescence was induced by treating the cells with 1 μg/mL LPS (Invivogen, San Diego, CA, USA) and 5 mM nicotine (Sigma-Aldrich, St. Louis, MO, USA) for 3 days. Following treatment, PDLCs were washed with Dulbecco’s PBS and incubated in serum-free medium for 24 h. The collected supernatant was filtered (0.2 μm) and mixed 1:1 with osteoclastogenic medium for subsequent experiments with BMMs.

### 4.3. ATF3 Knockdown

ATF3 expression was silenced using ATF3-specific siRNA (10~30 pmol) transfected with Lipofectamine 3000 (Thermo Fisher Scientific, Waltham, MA, USA). Control cells received scrambled siRNA (Bioneer, Daejeon, Republic of Korea). After 24 h, cells were exposed to LPS and nicotine, as described above.

### 4.4. Assessment of SASP Factors

CM was used to measure the quantification of NO and PGE_2_ secreted from the control and experimental groups of PDLCs. A kit was purchased, NO (abcam, Cambridge, UK) and PGE_2_ (R&D Systems, Minneapolis, MN, USA), and the experiment was conducted according to the company’s instructions. Fluorescence was measured using a fluorescence spectrophotometer (BioTek, Winooski, VT, USA), with excitation and emission at 540 nm from NO and 450 nm from PGE_2_.

Total RNA was isolated from cells using TRIzol reagent (Invitrogen, Carlsbad, CA, USA) according to the manufacturer’s instructions. Reverse transcription was performed with AccuPower RT PreMix (Bioneer, Daejeon, Republic of Korea). The resulting cDNA was amplified with AccuPower PCR PreMix (Bioneer). PCR products were resolved on 1.5% agarose gels and stained with ethidium bromide (Invitrogen). mRNA gel images were captured using the ChemiDoc™ XRS Imaging System (Bio-Rad Co., Hercules, CA, USA) and processed with Quantity-One ver 4.6 software (Bio-Rad). Each band was quantified using the ImageJ ver 1.54k software followed by statistical analysis. qPCR was performed on cDNA samples using SYBR Green PCR Master Mix (Roche Diagnostics, Basel, Switzerland) on a LightCycler (Roche Diagnostics). The relative mRNA levels of target genes were normalized to β-actin mRNA levels and analyzed using the comparative Ct method (ΔΔCt). The sequences of the specific primers used in this study, which were purchased from [Bioneer], are provided in Table 1.

### 4.5. Osteoclast Differentiation and Bone Resorption Assay

BMMs were isolated from 6-week-old male ICR mice (Institute of Cancer Research, Charles River Laboratories, Seoul, Republic of Korea) and used as precursor cells [35]. BMMs were seeded in culture plates and cultured in the presence of 30 ng/mL macrophage colony-stimulating factor (M-CSF; Peprotech, Cranbury, NJ, USA) and 100 ng/mL RANKL (Peprotech) in the presence or absence of CM (mixed at a 1:1 ratio) for 2–4 days. Osteoclasts were identified by tartrate-resistant acid phosphatase (TRAP) staining using an acid phosphatase kit (Sigma-Aldrich, St. Louis, MO, USA) following the manufacturer’s instructions, and bone resorption was quantified using dentin slice assays. BMMs on dentin slices were treated CM for 7 days. Cells were removed by wiping with a cotton swab, and dentin slices (IDS Immunodiagnostic systems, Tyne & Wear, UK) were stained with hematoxylin. Resorption pits were observed under a light microscope, and the resorption area was analyzed with Osteo-Measure^TM^ (OsteoMetrics7 ver 4.3.0.1, Inc., Atlanta, GA, USA). Representative data were obtained from three independent experiments.

### 4.6. Western Blotting

Cells were lysed in PRO-PREP protein extraction solution (Intron, Seoul, Republic of Korea) to prepare whole-cell lysates. Nuclear extracts were prepared from cultured cells using Pierce™ NE-PER™ Nuclear and Cytoplasmic Extraction Reagents (Thermo Fisher Scientific). Cells were harvested, washed, and resuspended in cytoplasmic extraction buffer, followed by incubation on ice for 5–10 min and centrifugation to separate the cytoplasmic and nuclear fractions. The nuclear pellet was resuspended in Pierce™ Nuclear Extraction Reagents and incubated on ice for 20–30 min to extract nuclear proteins, followed by centrifugation to collect the nuclear extract. Proteins (20~40 µg per lane) were separated by sodium dodecyl sulfate–polyacrylamide gel electrophoresis (SDS-PAGE) and transferred to polyvinylidene fluoride (PVDF) membranes. After transfer, membranes were washed three times with TBS-T then blocked for 1 h. Subsequently, membranes were incubated overnight with primary antibodies and then with horseradish peroxidase-conjugated secondary antibodies for 1 h at 37 °C. Primary antibodies against β-actin, p65, c-Jun, c-Fos, and Lamin B (Santa Cruz Biotechnology, Santa Cruz, CA, USA). As well as ATF3, phosphorylated ERK and STAT3 (Cell Signaling Technology, Beverly, MA, USA) were used at a dilution of 1:1000. Proteins were visualized using an enhanced chemiluminescence system (Amersham, Piscataway, NJ, USA). Images were captured with a ChemiDoc™ XRS Imaging System (Bio-Rad) and processed using Quantity-One software. Representative data were obtained from three independent experiments.

### 4.7. Immunofluorescence Assay

PDLCs were seeded onto coverslips at a density of 1 × 10^5^ cells/mL and cultured. Cells were fixed in 10% formalin for 15 min at room temperature. After three washes with phosphate-buffered saline (PBS), the cells were permeabilized with 0.2% Triton X-100 in PBS for 20 min, washed three times with PBS, and then blocked with 5% bovine serum albumin in PBS for 1 h at room temperature. Cells were incubated overnight at room temperature with primary antibodies against p-STAT3, p65, and c-Jun (1:200), washed three times, and then incubated with Alexa-488-conjugated secondary antibodies (1:500; Invitrogen) for 2 h at room temperature. The nuclei were stained with propidium iodide (PI; Sigma-Aldrich) for 5 min. After rinsing three times, coverslips were mounted on glass slides and examined using a confocal microscope (Cell Voyager, Yokohama, Japan). Representative data were obtained from three independent experiments.

### 4.8. Statistical Analysis

Data are expressed as the mean ± standard deviation, and statistical analyses were performed using GraphPad Prism 8.4.3 software (San Diego, CA, USA). Statistical significance between groups was evaluated by *t*-test of variance (ANOVA) followed by Dunnett’s multiple comparison tests, and *p* < 0.05 was considered significant. *** stands for *p* < 0.001; ** stands for *p* < 0.01; * stands for *p* < 0.05. ns; not significant.

## 5. Conclusions

Our study highlights the critical role of ATF3 in PDLC senescence and osteoclastogenesis. ATF3 inhibition reduced inflammatory SASP factors, decreased the RANKL/OPG ratio, and suppressed osteoclast differentiation. Mechanistically, ATF3 regulated these processes via NF-κB, AP-1, STAT3, and ERK pathways. These findings suggest that targeting ATF3 could provide a novel therapeutic strategy for age-related periodontal diseases. Future research should further explore ATF3’s regulatory mechanisms and its broader impact on aging and bone metabolism.

## Figures and Tables

**Figure 1 ijms-26-04959-f001:**
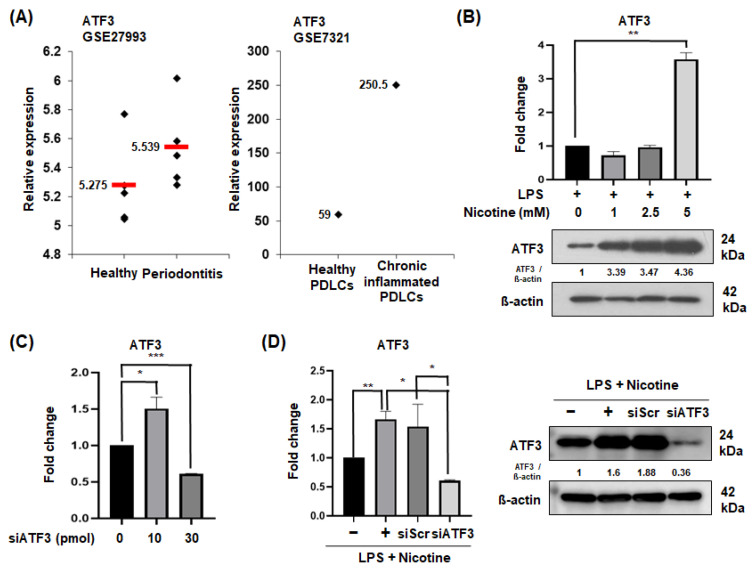
Expression of ATF3 mRNA and protein. (**A**) Analysis of the data from the GEO dataset. GSE27993 provides ATF3 gene expression data from human periodontal ligament (PDL) tissues of healthy individuals and periodontitis patients (left). GSE7321 focuses on ATF3 gene expression in human PDLCs from healthy and periodontitis-affected individuals (right). (**B**) Dose-dependent expression of ATF3 mRNA or protein in PDLCs treated with LPS (1 μg/mL) plus nicotine (5 mM) for 3 days. (**C**,**D**) Effective knockdown of ATF3 by siRNA. Treatment with ATF3 and Scr siRNA for 1 day, followed by treatment with LPS (1 μg/mL) and nicotine (5 mM) for 3 days. Data are representative of three independent experiments. All data are represented as the mean ± SD. Statistically significant differences were indicated by asterisks in the graphs, with * *p* < 0.05, ** *p* < 0.01, and *** *p* < 0.001.

**Figure 2 ijms-26-04959-f002:**
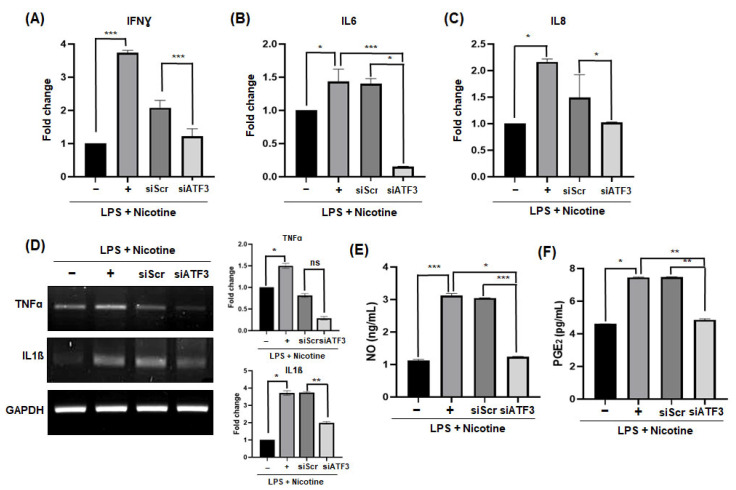
Influence of ATF3 on SASP factors induced by LPS and nicotine in PDLCs. PDLCs were transfected with ATF3 (30 pmol) and Scr (30 pmol) siRNA, followed by treatment with LPS (1 μg/mL) and nicotine (5 mM) for 3 days. (**A**–**C**) Gene expression levels of IFNγ, IL6, and IL8 were assessed by qPCR. (**D**) Gene expression of TNFα and IL1β was assessed by RT-PCR. The production of (**E**) NO and (**F**) PGE_2_ was measured using respective kits. Data are representative of three independent experiments. All data are represented as the mean ± SD. Statistically significant differences were indicated by asterisks in the graphs, with * *p* < 0.05, ** *p* < 0.01, and *** *p* < 0.001.

**Figure 3 ijms-26-04959-f003:**
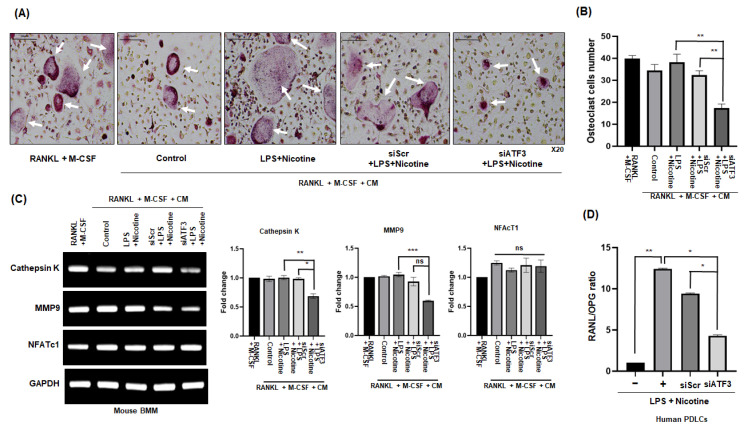
Impact of ATF3 on LPS and nicotine-induced osteoclastogenesis in mouse BMMs. Human PDLCs were pretreated with either scramble (Scr) or ATF3 siRNA for 24 h, then incubated with 1 μg/mL LPS and 5 mM nicotine for 3 days, then changed to serum-free media, and CM was collected the next day. (**A**) The BMM cells were incubated with media containing M-CSF and RANKL, with or without CM. After 4 days of culture, the cells were fixed, and osteoclast-like cells were identified by TRAP staining and (**B**) quantified based on cells with three or more nuclei. White arrows point to cells stained positive for TRAP. (**C**) Expression of osteoclast-specific marker genes was assessed in BMM cells by RT-PCR. (**D**) The expression ratio of RANKL to OPG in senescence-induced PDLCs with or without ATF3 inhibition was measured using ΔΔCt values obtained from qPCR. Data are representative of three independent experiments. All data are represented as the mean ± SD. Statistically significant differences were indicated by asterisks in the graphs, with * *p* < 0.05, ** *p* < 0.01, and *** *p* < 0.001. ns: not significant. Scale bar = 50 μm.

**Figure 4 ijms-26-04959-f004:**
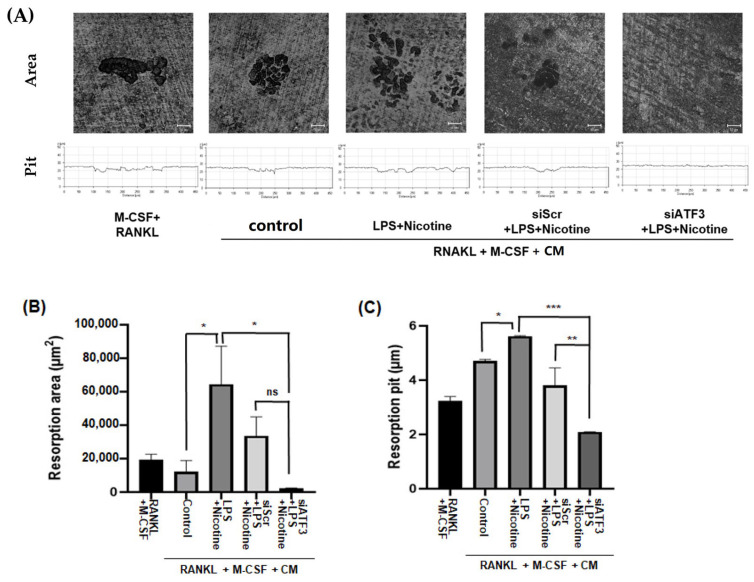
Role of ATF3 in LPS and nicotine-induced bone resorption. Human PDLCs were pretreated with either scramble (Scr) or ATF3 siRNA for 24 h, then incubated with 1 μg/mL LPS and 5 mM nicotine for 3 days. Afterward, the cells were switched to serum-free media, and conditioned media (CM) were collected the next day. (**A**) BMM cells were cultured on dentin slides with media containing M-CSF and RANKL, with or without CM. After 1 week of incubation, cells on the dentin slides were removed, and images were captured. The images were analyzed using OsteoMeasure7 Ver 4.3.0.1 software to quantify (**B**) the number of resorption area and (**C**) their pits. Data represent three independent experiments. Statistically significant differences were observed among the groups treated with LPS and nicotine, groups treated with Scr siRNA followed by LPS and nicotine, and groups treated with ATF3 siRNA followed by LPS and nicotine. * *p* < 0.05, ** *p* < 0.01, and *** *p* < 0.001. ns: not significant. Scale bar = 10 μm.

**Figure 5 ijms-26-04959-f005:**
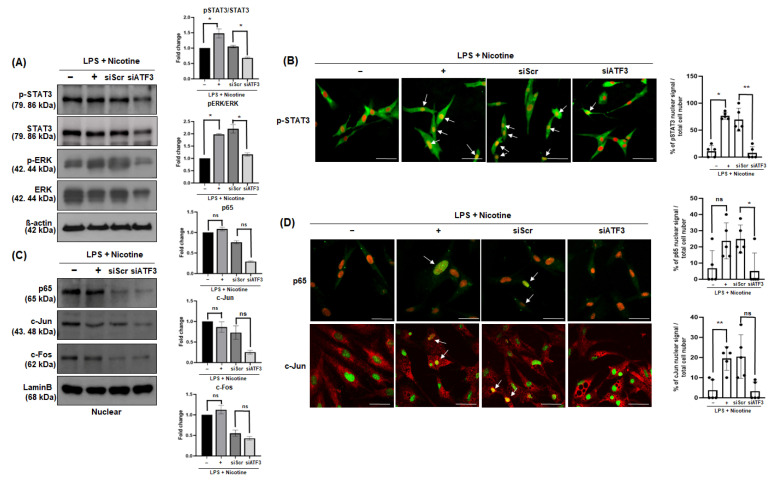
Senescence-inducing signals in PDLCs and the role of ATF3. (**A**–**D**) PDLCs were transfected with ATF3 (30 pmol) and Scr (30 pmol) siRNA, followed by treatment with LPS (1 μg/mL) and nicotine (5 mM). The expression levels of STAT3, ERK, p65, and AP-1 (c-Jun and c-Fos) were analyzed by Western blot (**A**,**C**) and immunocytochemistry (**B**,**D**) after 30 min (**A**,**C**) and 60 min (**B**,**D**) of treatment. A change in the color of nuclei from red to yellow (due to co-localization of the green fluorescein isothiocyanate fluorescence signal and the red propidium iodide fluorescence signal, indicated by arrows) was indicative of STAT3, p65, and c-Jun translocation in the cells. Data are presented as the percentage of cells showing nuclear signal relative to the total number of cells in each field. Results are representative of three independent experiments. * *p* < 0.05, ** *p* < 0.01. ns: not significant. Scale bar = 200 μm.

**Table 1 ijms-26-04959-t001:** Primers sequences for PCR.

Primer	Sequences	Tm	Size
qPCR primer			
Human β-actin	Forward 5′-CACCATTGGCAATGAGCGGTTC-3′ Reverse 5′-AGGTCTTTGCGGATGTCCACGT-3′	60	135
Human IFN-γ	Forward 5′-TGTCGCCAGCAGCTAAAACA-3′ Reverse 5′-TGCAGGCAGGACAACCATTA-3′	60	91
Human IL6	Forward 5′-CACCGGGAACGAAAGAGAAGC-3′ Reverse 5′-CGAAGGCGCTTGTGGAGA-3′	60	75
Human IL8	Forward 5′-CCACCGGAGCACTCCATAAG-3′ Reverse 5′-GATGGTTCCTTCCGGTGGTT-3′	60	97
Human ATF3	Forward 5′-GTTTGAGGATTTTGCTAACCTGAC-3′ Reverse 5′-AGCTGCAATCTTATTTCTTTCTCG-3′	55	211
Human RANKL	Forward 5′-AAAGCCGGGCTCCAAGTC-3′ Reverse 5′-TTCTTGTCTGCGGCCAACTC-3′	60	72
Human OPG	Forward 5′-CCTGGCACCAAAGTAAACGC-3′ Reverse 5′-GCACGCTGTTTTCACAGAGG-3′	60	163
RT-PCR primer			
Human GAPDH	Forward 5′-CTCTTCACCACCATGGAGAAG -3′ Reverse 5′-GTTGTCATGGATGACCTTGGC -3′	58	201
Human TNF-α	Forward 5′-GGAAGACCCCTCCCAGATAG-3′ Reverse 5′-CCCCAGGGACCTCTCTCTAA-3′	52	413
Human IL1β	Forward 5′-GGATATGGAGCAACAAGTGG-3′ Reverse 5′-ATGTACCAGTTGGGGAACTG-3′	60	264
Mouse GAPDH	Forward 5′-GAGAGTGTTTCCTCGTCCCG-3′ Reverse 5′-ACTGTGCCGTTGAATTTGCC-3′	60	201
Mouse Cathepsin K	Forward 5′-GCCACGCTTCCTATCCGAAA-3′ Reverse 5′-AGCTGAAAGCCCAACAGGAA-3′	60	481
Mouse MMP-9	Forward 5′-AACCTCCAACCTCACGGACA-3′ Reverse 5′-CGCGGCAAGTCTTCAGAGTA-3′	65	294
Mouse NFATc1	Forward 5′-GGTAACTCTGTCTTTCTAACCTTAAGCTC-3′ Reverse 5′-GTGATGACCCCAGCATGCACCAGTCACAG-3′	60	240

## Data Availability

The data presented in this study are available on request from the authors.

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
