# Peer review of "Activating Transcription Factor 3 (ATF3) Regulates Cellular Senescence and Osteoclastogenesis via STAT3/ERK and p65/AP-1 Pathways in Human Periodontal Ligament Cells"

_ijms, 2025, doi:10.3390/ijms26104959_

Round 1
Reviewer 1 Report
Comments and Suggestions for Authors
1、Please offer the ethical approvals for the use of primary human PDLCs.
2、Please offer the reference for the chosen concentration of LPS (1 μg/mL) and nicotine (5 μM) to induce senescence in PDLCs. Besides, the reason for the time spot should be explained.
3、The authors propose ATF3 as a therapeutic target. However, the author also mentioned ATF3 has dual roles (pro-survival vs. pro-apoptotic) depending on cellular context. Please give some evidence about potential off-target effects or unintended consequences of ATF3 inhibition in periodontal tissues?
4、Please mention the biological replicates for the biological experiments in Materials and Methods.
I think it's okay for readers.
Author Response
Thank you very much for taking the time to review this manuscript. Please find the detailed responses below and the corresponding revisions/corrections highlighted/in track changes in the re-submitted files.
Comments 1: Please offer the ethical approvals for the use of primary human PDLCs.
Response 1: Thank you for your meaningful comments. Human periodontal ligament cells (PDLCs) were purchased from ScienCell Research Laboratories (catalog #2630, Carlsbad, CA, USA). As these cells were obtained from a commercial source that adheres to ethical procurement guidelines, no additional institutional ethical approval was required for their use in this study [page 9].
Comments 2: Please offer the reference for the chosen concentration of LPS (1 μg/mL) and nicotine (5 μM) to induce senescence in PDLCs. Besides, the reason for the time spot should be explained.
Response 2: Agree. We have, accordingly, modified to emphasize this point [page 7]
Our previously reported study examined the inflammatory response by treating PDLCs with LPS and nicotine, and there are several papers that tested LPS 1ug/ml and nicotine at different concentrations (e.g., *Nicotine and lipopolysaccharide stimulate the production of MMPs and prostaglandin E2 by hypoxia-inducible factor-1α up-regulation in human periodontal ligament cells* — J Periodontal Res. 2012 Dec;47(6):719-28. doi: 10.1111/j.1600-0765.2012.01487.x; *PIN1 Inhibition Suppresses Osteoclast Differentiation and Inflammatory Responses* — J Dent Res. 2016 Nov;95(12):1415-1424. doi: 10.1177/0022034516659642; *HIF-2 Inhibition Suppresses Inflammatory Responses and Osteoclastic Differentiation in Human Periodontal Ligament Cells* — J Cell Biochem. 2015 Jul;116(7):1241-55. doi: 10.1002/jcb.25078). although the treatment time was different (previous experiment was treated for 24 h, current experiment is treated for 72 h), major factors related to aging, COX-2, PGE2. It has been confirmed that the expression of iNOS, NO, TNFa, IL1b, IL6, and IL8 increases. In addition, there was a research report that stress-induced premature senescence occurs in normal human fibroblasts, which are somatic cells, when stress-inducing substances such as hydrogen peroxide (H2O2) or EtOH are treated for 72 h at a concentration that does not cause cytotoxicity. In fact, there is a previous research result that aging is promoted when EtOH is treated for 72 h in periodontal ligament cells (e.g., *Effects of Melatonin and Its Underlying Mechanism on Ethanol-Stimulated Senescence and Osteoclastic Differentiation in Human Periodontal Ligament Cells and Cementoblasts* — Int J Mol Sci. 2018 Jun 12;19(6):1742. doi: 10.3390/ijms19061742.). Although the data was not shown, the concentration and time were determined based on previous experimental results and experimental studies that treated LPS and nicoine.
Comments 3: The authors propose ATF3 as a therapeutic target. However, the author also mentioned ATF3 has dual roles (pro-survival vs. pro-apoptotic) depending on cellular context. Please give some evidence about potential off-target effects or unintended consequences of ATF3 inhibition in periodontal tissues?
Response 3: Thank you for your insightful comments.
Recent studies have demonstrated that ATF3 promotes ferroptosis and inhibits osteogenic differentiation in LPS-induced inflammatory human periodontal ligament stem cells (hPDLSCs). Suppression of ATF3 has been shown to activate the Nrf2/HO-1 pathway, thereby reducing inflammation and ferroptosis while enhancing osteogenic differentiation (e.g., *ATF3 affects osteogenic differentiation in inflammatory hPDLSCs by mediating ferroptosis via regulating the Nrf2/HO-1 signaling pathway* — Tissue and Cell, 89, 102447. https://doi.org/10.1016/j.tice.2024.102447). However, research on the role of ATF3 in regulating cell survival and apoptosis within periodontal tissues remains limited. Inhibition of ATF3 in periodontal tissues requires careful balancing of these opposing effects. While it may reduce harmful inflammation and promote osteogenesis in diseased cells, it could concurrently impair normal cell survival, wound healing, and immune regulation in the periodontium. Thus, precise targeting strategies—such as cell-type-specific approaches—and vigilant monitoring of potential off-target effects will be critical in any therapeutic applications involving ATF3 suppression. Further investigation is needed to fully elucidate the biological functions of ATF3 in the context of periodontal health and disease.
Comments 4: Please mention the biological replicates for the biological experiments in Materials and Methods.
Response 4: According to your comments, we revised it [Materials and Methods].
“Representative data were obtained from three independent experiments.”
- Response to Comments on the Quality of English Language
Point 1: I think it's okay for readers.
Response 1: Thank you for your kind comment. We appreciate your positive feedback regarding the readability and clarity of the manuscript.

Reviewer 2 Report
Comments and Suggestions for Authors
please see enclosed pdf

Author Response
Thank you very much for taking the time to review this manuscript. Please find the detailed responses below and the corresponding revisions/corrections highlighted/in track changes in the re-submitted files.
Comments 1: The authors should also mention the effect of bacteria on the oral cavity and its involvement on the pathogenesis of periodontitis. I suggest:
Solomon, S.M.; Matei, M.; Badescu, A.C.; Jelihovschi, I.; Martu-Stefanache, A.; Teusan, A.; Martu, S.; Iancu, L.S. Evaluation of DNA extraction methods from saliva as a source of PCR-amplifiable genomic DNA. Rev. Chim. 2015, 67, 2101–2103.
Response 1: Thank you for pointing this out. I/We agree with this comment. Therefore, we have revised the addition of “In addition to host factors, the oral microbiota plays a critical role in the initiation and progression of periodontitis, as bacterial colonization and biofilm formation trigger chronic inflammation and tissue destruction. The significance of bacterial involvement in the oral cavity has been widely acknowledged in the context of periodontal disease pathogenesis [4].” [line 39]
Ref [4]: Feres, M.; Teles, F.; Teles, R.; Figueiredo, L.C.; Faveri, M.; The subgingival periodontal microbiota of the aging mouth. Periodontol 2000. 2016, 72, 30–53. doi.org/10.1111/prd.12136.
Comments 2: The images are too small and authors should use arrows to better exemplify histological findings
Response 2: According to your comments, we revised it [page 5, Figure 3].
Comments 3: make images larger
Response 3: According to your comments, we revised it [page 7, Figure 5].
Comments 4: The discussions are too small, please expand
Response 4: Thank you for your kindful comments. According to your comments, we revised in manuscripts [page 7].
Comments 5: the authors should expand on the roles of MMP in periodontitis and its effect on bone loss
Response 5: Thank you for pointing this out. According to your comments, we revised it [Line234]
“MMPs (matrix metalloproteinases) degrade collagen and extracellular matrix in periodontitis, inducing the destruction of gingival connective tissue and alveolar bone. In particular, MMP-8 and MMP-9 are deeply involved in inflammatory responses and bone loss, and also affect osteoclast activation. MMP levels are high in the saliva and gingival crevicular fluid of periodontitis patients, and this is associated with the severity of the disease, drawing attention as a biomarker and therapeutic target [27].”

Reviewer 3 Report
Comments and Suggestions for Authors
In the present manuscript, the authors study the role of activating transcription factor 3 (ATF3) in the regulation of periodontal ligament cell (PDLC) senescence and osteoclastogenesis in mouse bone marrow macrophages (BMMs). The role of environmental stressors like lipo-polysaccharide (LPS) and nicotine, which induce oxidative stress and inflammation and contribute to oral cellular senescence, has been studied. The authors proved that ATF3 silencing in PDLCs reduced the LPS/nicotine-induced senescence. Moreover, they demonstrated that the condition media (CM), derived from ATF3-silenced PDLCs which were induced to senescence, inhibited the differentiation of BMMs into osteoclasts. The study contributes to the understanding of the role of ATF3 in a senescent microenvironment with interesting implications in the fight of its consequences, the development of periodontal disease and alveolar bone loss.
Major comments:
Periodontal disease is a chronic inflammatory disease caused by bacteria accumulated in the gingival sulcus and periodontal pocket that induces an inflammatory immune response. Smoking is a major risk factor for periodontal disease as nicotine accumulates in periodontal tissue. The authors demonstrated that ATF3 gene expression and protein abundance were increased upon LPS- and nicotine-mediated induced senescence.
- In the sentence at lines 69-71 “When senescence was induced in a dose-dependent manner with LPS and nicotine, ATF3 expression gradually increased, with a significant rise in protein levels at 5 µM nicotine (Figure 1B)”, it should be specified that the increment is visible at the protein level in a dose-dependent manner since it increases only at 5 µM nicotine at the mRNA level.
- May the authors quantify the ATF3 protein abundance in Figure 1B?
- Which statistical test has been used in the graph in Figure 1B? Which samples have been compared? Could you please specify which comparison the ** refer to? Could the authors describe properly the graph in the legend by mentioning all the different concentrations of nicotine used?
- May the authors compare the effect of LPS only, nicotine only and LPS+nicotine treatment in PDLCs on ATF3 gene expression and protein abundance? It would help to understand how ATF3 is influenced by the different treatments.
- May the authors evaluate the cell viability of PDLCs during LPS and nicotine treatment at various concentrations?
- In relation to the sentence at the lines 72-74 (“To inhibit ATF3 expression induced by LPS and nicotine, siRNA was used, and both 20 nM and 30 nM concentrations similarly reduced expression (Figure 1C)”), the concentration of siRNA mentioned is of 20 nM and 30nM while in Figure 1C 10 pmol and 30 pmol are used. In the Material and Methods, the concentration reported is of 20 nM (line 268). Please clarify the concentration of siATF3 used for the graph in Figure 1C and write it with the same unit (nM) used in the main text. Could you please specify which comparison the * refer to?
- The Scr sample constitutes a negative control as the RNA does not anneal with any specific sequence in order to silence the expression of a gene, therefore it is possible to expect that this sample has a similar behaviour to the PDLCs grown in LPS/nicotine-induced senescence environment (+) sample. Nevertheless, this does not occur in a few experiments (2A, 4C). This can be due to the silencing treatment or other biological or technical issues. I suggest to not compare the PDLCs grown in LPS/nicotine-induced senescence environment (+) sample to ATF3 siRNA sample, but to compare the PDLCs grown in control environment (-) with the PDLCs grown in LPS/nicotine-induced senescence environment (+) samples, and the PDLCs grown in LPS/nicotine-induced senescence environment treated with scr RNA with the PDLCs grown in LPS/nicotine-induced senescence environment treated with ATF3 siRNA samples separately as two independent comparisons.
- In the graph in Figure 1D, could the authors compare statistically the - vs + samples and the Scr vs SiATF3 samples? May the authors quantify the ATF3 protein abundance in Figure 1D?
- Please revise the Figure 1 legend. The description is not complete and the information is incomplete/incorrect.
The authors proved that ATF3 regulates the expression of cellular senescence phenotype (SASP) by releasing several SASP factors, such as IFNɣ, IL6 and IL8 as shown in Figure 2A-C (mentioned as TNFα, IL1β, and IL6 in the main text in relation to Figure 2A at lines 85-87; please correct with the right names of factors throughout the text, included the discussion part).
- May the authors compare statistically the production/secretion of SASP factors from PDLCs grown in control environment (-) and PDLCs grown in LPS/nicotine-induced senescence environment (+) samples in all graphs in Figure 2?
- May the authors compare statistically the production/secretion of SASP factors from PDLCs grown in LPS/nicotine-induced senescence environment and treated with scr RNA with PDLCs grown in LPS/nicotine-induced senescence environment and treated with ATF3 siRNA samples in all graphs in Figure 2?
- Line 92: please correct the number of Figure to Figure 2E-F.
- Please revise the Figure 2 legend. The description is not complete and the information is incomplete/incorrect; some p values are missing.
- Can the authors prove and support these findings on the role of ATF3 in senescence by evaluating other aspects and performing other assays to identify aged cells, such as the senescence-associated beta-galactosidase (SA-βgal) activity? Is there other osteoclastogenic cytokines or other SASP factors that could be evaluated?
- May the authors evaluate the production of SASP factors in relation to nicotine concentration? May the authors determine whether the process is nicotine concentration-dependent? Is it aligned with ATF3 expression upon nicotine-concentration treatment?
- Can the authors test the production of SASP factors when PDLCs are treated with nicotine only? Can the nicotine only treatment be compared with the nicotine + LPS treatment?
Nicotine and LPS can stimulate PI3K, MAPK and NF-kB pathways which could determine the expression of SASP factors in senescent cells. The authors state that ATF3 knockdown inhibits STAT3 and ERK phosphorylation and p65 and c-Jun translocation in PDLCs. The authors should improve the quality of the WB results and relative quantifications in order to support their statements. Further studies on the involvement of these and other signaling molecules and relative pathways in the regulation of ATF3-mediated induction of senescence, inflammation, periodontitis and bone resorption should be performed.
- Figure 5A: I suggest to replace the blots where the signal might be over-exposed (STAT3, beta-actin) with blots with a lower signal exposure. Signals should be quantified and p-STAT3/STAT3 and p-ERK/ERK ratios should be calculated.
- Please make images in fig 5B and 5D bigger in size and with a magnification of 40X as it can be difficult to appreciate the details. Quantification of the number of cells with nuclear signal relative to the total number of cells should be displayed.
- Figure 5C: I suggest to replace the lamin B blot as the signal is not comparable in all samples. Please replace all the blots relative to the same experiment. Signals should be quantified and normalized on the loading control.
- Please add molecular weights to all blots.
The authors prove that the silencing of ATF3 in PDLCs determines the formation of a lower number of osteoclasts. In line with this, an increase in osteoclastogenesis would be expected in case of overexpression of ATF3, which would be induced by the treatment with LPS and nicotine. Nevertheless, a comparison between BMMs treated with RANKL + M-CSF + CM in control environment (-) and in LPS/nicotine-induced senescence environment (+) does not seem to show this effect.
- May the authors explain how ATF3 could behave differently?
- Figure 3B: Could the authors compare statistically the PDLCs grown in control environment (-) and PDLCs grown in LPS/nicotine-induced senescence environment (+) samples?
In addition, the expression of genes involved in osteoclast differentiation, such as Cathepsin K, MMP-9 and NFATc1, is not affected.
- May the authors explain these results? May the authors explain how ATF3 can induce osteoclastogenesis if the genes involved in osteoclast differentiation are not downregulated upon its silencing, or upregulated upon ATF3 overexpression?
- Can the authors replace Con with – or control in the graph in Figure 3C?
- Do the authors know the effect of ATF3 silencing on Acp5, c-Fos, tartrate resistant acid phosphate (TRAP), osteoclast-associated receptor (OSCAR) and the d2 isoform of vacuolar ATPase V(0) domain gene expression in BMMs? Can the authors assess osteoclastogenesis by evaluating TRAP activity?
I would like to suggest the authors to investigate deeper the role of ATF3 in PDLCs in LPS/nicotine-induced senescence environment in relation to the differentiation of BMMs into osteoclasts. May the authors provide other findings to support their statement?
Do the authors know whether the lower RANKL/OPG ratio in ATF3-silenced PDLCs can be related to a lower abundance of RANKL in in ATF3-silenced PDLCs compared to control cells? Do the authors know whether the effect of PDLCs CM on BMMs differentiation into osteoclasts may be mediated by RANKL? Could they investigate this aspect? RANKL can bind to RANK on the plasmatic membrane of BMMs. RANK activation leads to increased intracellular Ca2+, activation of the c-Fos and NFATs transcription factors that regulate osteoclast-specific gene transcription, their association with ATF3, transcriptional activity and osteoclast differentiation.
- Figure 4: could the authors compare statistically the PDLCs grown in control environment (-) and PDLCs grown in LPS/nicotine-induced senescence environment (+) samples?
- Figure 4B: please explain the quantification indicated as “resorption pit”.
The authors prove a reduction of SASP factors upon ATF3 depletion in PDLCs. These factors are components of CM which is used by the authors to induce osteoclastogenesis of BMMs. It might be interesting to discuss how specific SASP factors delivered directly on BMMs could improve osteoclastogenesis. This could support the hypothesis that the role of ATF3 in osteoclastogenesis is mediated by SASP factors.
Minor comments:
Line 45: the word periodontitis is repeated twice.
Figure 1: units in the graphs for ATF3 mRNA expression are missing.
Figure 1A legend: legend is not complete.
Figure 2A-C: units (y axe) are missing.
Lines 106-110: please revise for clarity.
Line 115: MMP9.
Figure 3A: the names of the samples are incorrect. The text is misleading.
Figure 3B: replace RNAKL with RANKL.
Figure 3D: unit (y axe) is missing.
Figure 3: please align text with graphs.
Figure 3 legend at line 133 “Human PDLCs were pretreated with ATF3 for 24 h”: please clearly refer to Scr and ATF3 siRNA treatment.
Figure 4: replace RNAKL with RANKL.
Figure 4a legend at line 146: please replace ATF3 treatment with “Human PDLCs were treated with either Scr or ATF3 siRNA for 24h, …”.
Line 159: please check the sentence “Following the silencing ATF3, …”.
Lines 164-166: please check the sentence.
Line 191: please check the sentence.
Lines 235-238: the sentence is not clear.
Line 257: G is missing in the text (in GSE7321).
Lines 288-289: please specify the company where the primers have been obtained.
Line 292: please explain acronym “ICR mice”.
Line 300: please check the sentence.
Line 307: please describe the protocol for preparing nuclear extracts.
Line 333: please explain all p values; **** stands for p < 0.0001; *** stands for p<0.001; ** stands for p<0.01; * stands for p<0.05.
Lines 424-427: references 34 and 35 are the same.
The authors aim at identifying the mechanisms associated with the regulation of SAPS, osteoclast differentiation, and activity in senescence-induced PDLCs with controlled ATF3 expression. In order to reveal the molecular mechanisms, I suggest the authors to improve the quality of the data, to support and corroborate the present findings as suggested and to include the characterization of the associated signaling pathways.
Author Response
Thank you very much for taking the time to review this manuscript. Please find the detailed responses below and the corresponding revisions/corrections highlighted/in track changes in the re-submitted files.
Major comments:
Comments 1: In the sentence at lines 69-71 “When senescence was induced in a dose-dependent manner with LPS and nicotine, ATF3 expression gradually increased, with a significant rise in protein levels at 5 µM nicotine (Figure 1B)”, it should be specified that the increment is visible at the protein level in a dose-dependent manner since it increases only at 5 µM nicotine at the mRNA level.
May the authors quantify the ATF3 protein abundance in Figure 1B?
Response 1: Thank you for pointing this out. According to your comments, we revised it [Line 72].
“When senescence was induced in a dose-dependent manner with LPS and nicotine, a significant increase in ATF3 protein levels at 5 mM nicotine (Figure 1B).”
Comments 2: Which statistical test has been used in the graph in Figure 1B? Which samples have been compared? Could you please specify which comparison the ** refer to? Could the authors describe properly the graph in the legend by mentioning all the different concentrations of nicotine used?
Response 2: According to your comments, we revised it. All protein blots were quantified using ImageJ, and an unpaired t-test was conducted to compare the results with a control group that did not receive nicotine treatment (Figure 1B).
Comments 3: Could the authors describe properly the graph in the legend by mentioning all the different concentrations of nicotine used? May the authors compare the effect of LPS only, nicotine only and LPS+nicotine treatment in PDLCs on ATF3 gene expression and protein abundance? It would help to understand how ATF3 is influenced by the different treatments.
Response 3: Thank you for your kindful comments. The combination of LPS and various concentrations of nicotine is commonly used in vitro to mimic the inflammatory and degenerative microenvironment observed in periodontal disease. This model allows for the study of cellular responses in an environment that closely resembles chronic periodontitis. Although we did not present results for LPS or nicotine treatment alone, Chang et al. reported that co-stimulation with LPS and nicotine in periodontal ligament cells (PDLCs) significantly upregulated stress response genes, such as activating transcription factor 3 (ATF3) [12].
Comments 4: May the authors evaluate the cell viability of PDLCs during LPS and nicotine treatment at various concentrations?
Response 4: Thank you for your valuable comment. Although the data are not shown in the manuscript, we confirmed that treatment with various concentrations of LPS and nicotine did not cause significant cytotoxicity in PDLCs, as assessed by cell viability assays. Therefore, we proceeded with the subsequent experiments based on these preliminary findings.
Comments 5: In relation to the sentence at the lines 72-74 (“To inhibit ATF3 expression induced by LPS and nicotine, siRNA was used, and both 20 nM and 30 nM concentrations similarly reduced expression (Figure 1C)”), the concentration of siRNA mentioned is of 20 nM and 30nM while in Figure 1C 10 pmol and 30 pmol are used. In the Material and Methods, the concentration reported is of 20 nM (line 268). Please clarify the concentration of siATF3 used for the graph in Figure 1C and write it with the same unit (nM) used in the main text. Could you please specify which comparison the * refer to?
Response 5: According to your comments, we revised in figure and manuscripts [Line 75-78, Figure 1].
Comments 6: The Scr sample constitutes a negative control as the RNA does not anneal with any specific sequence in order to silence the expression of a gene, therefore it is possible to expect that this sample has a similar behaviour to the PDLCs grown in LPS/nicotine-induced senescence environment (+) sample. Nevertheless, this does not occur in a few experiments (2A, 4C). This can be due to the silencing treatment or other biological or technical issues. I suggest to not compare the PDLCs grown in LPS/nicotine-induced senescence environment (+) sample to ATF3 siRNA sample, but to compare the PDLCs grown in control environment (-) with the PDLCs grown in LPS/nicotine-induced senescence environment (+) samples, and the PDLCs grown in LPS/nicotine-induced senescence environment treated with scr RNA with the PDLCs grown in LPS/nicotine-induced senescence environment treated with ATF3 siRNA samples separately as two independent comparisons.
In the graph in Figure 1D, could the authors compare statistically the - vs + samples and the Scr vs SiATF3 samples? May the authors quantify the ATF3 protein abundance in Figure 1D?
Response 6: Thank you for your constructive suggestions. As requested, we have now performed statistical comparisons between the (-) and (+) samples as well as between the Scr and SiATF3 samples in Figure 1D. In addition, we have quantified and presented the ATF3 protein abundance in Figure 1D. The figure and corresponding legend have been updated accordingly.
Comments 7: Please revise the Figure 1 legend. The description is not complete and the information is incomplete/incorrect.
Response 7: According to your comments, we revised it [Line 80-86].
Comments 8: The authors proved that ATF3 regulates the expression of cellular senescence phenotype (SASP) by releasing several SASP factors, such as IFNɣ, IL6 and IL8 as shown in Figure 2A-C (mentioned as TNFα, IL1β, and IL6 in the main text in relation to Figure 2A at lines 85-87; please correct with the right names of factors throughout the text, included the discussion part).
Response 8: According to your comments, we revised it [Line 91-97, 197, 212].
Comments 9: May the authors compare statistically the production/secretion of SASP factors from PDLCs grown in control environment (-) and PDLCs grown in LPS/nicotine-induced senescence environment (+) samples in all graphs in Figure 2?
May the authors compare statistically the production/secretion of SASP factors from PDLCs grown in LPS/nicotine-induced senescence environment and treated with scr RNA with PDLCs grown in LPS/nicotine-induced senescence environment and treated with ATF3 siRNA samples in all graphs in Figure 2?
Response 9: Thank you for your valuable comments.
In response, we have performed detailed statistical comparisons in all graphs of Figure 2 as follows:
We statistically compared the production and secretion levels of SASP factors between PDLCs cultured in a control environment (-) and those cultured in an LPS/nicotine-induced senescence environment (+). We also statistically compared the production and secretion levels of SASP factors between PDLCs grown under LPS/nicotine-induced senescence conditions and treated with Scr RNA and those treated with ATF3 siRNA. Statistically significant differences were clearly indicated, with p-values < 0.05 considered significant.
Comments 10: Line 92: please correct the number of Figure to Figure 2E-F.
Response 10: According to your comments, we revised it [Line 99].
Comments 11: Please revise the Figure 2 legend. The description is not complete and the information is incomplete/incorrect; some p values are missing.
Response 11: According to your comments, we revised it [page 4, Figure 2 legend]
Comments 12: Can the authors prove and support these findings on the role of ATF3 in senescence by evaluating other aspects and performing other assays to identify aged cells, such as the senescence-associated beta-galactosidase (SA-βgal) activity? Is there other osteoclastogenic cytokines or other SASP factors that could be evaluated?
Response 12: Thank you for your insightful suggestion. In our study, we quantified various SASP factors, such as IL-6 and IL-8, to more broadly assess the senescence status and specifically identified osteoclastogenic cytokines involved in aging and inflammatory conditions. TNFα, a major component of SASP, is a potent inflammatory cytokine that promotes osteoclastogenesis, while IL-1β also acts as a stimulator of osteoclastogenesis. Nevertheless, in addition to SA-β-gal staining—which detects senescent cells in senescence-induced PDLCs—protein expression analysis of key cell cycle inhibitors associated with senescence, such as p16^INK4a, p21^CIP1, and p53, as well as markers of DNA damage, may better identify characteristic features of senescent cells. We plan to include these analyses in subsequent studies.
Comments 13: May the authors evaluate the production of SASP factors in relation to nicotine concentration? May the authors determine whether the process is nicotine concentration-dependent? Is it aligned with ATF3 expression upon nicotine-concentration treatment?
Can the authors test the production of SASP factors when PDLCs are treated with nicotine only? Can the nicotine only treatment be compared with the nicotine + LPS treatment?
Response 13: Thank you for your kindful comments. Although the results for LPS or nicotine treatment alone were not presented, Chang et al. reported that co-stimulation with LPS and nicotine significantly upregulated stress response genes, such as activating transcription factor 3 (ATF3), in periodontal ligament cells (PDLCs) [12]. This model allows us to study cellular responses in an environment that closely resembles chronic periodontitis. Therefore, we focused on co-stimulation with LPS and nicotine in this study.
Comments 14: Nicotine and LPS can stimulate PI3K, MAPK and NF-kB pathways which could determine the expression of SASP factors in senescent cells. The authors state that ATF3 knockdown inhibits STAT3 and ERK phosphorylation and p65 and c-Jun translocation in PDLCs. The authors should improve the quality of the WB results and relative quantifications in order to support their statements. Further studies on the involvement of these and other signaling molecules and relative pathways in the regulation of ATF3-mediated induction of senescence, inflammation, periodontitis and bone resorption should be performed.
Response 14: Agree. Thank you for your insightful comments. Furthermore, our study demonstrates that ATF3 regulates STAT3 and ERK phosphorylation, and p65 and c-Jun translocation in PDLC aging and inflammation, but further studies are needed to elucidate the broader signaling mechanisms involved. We believe that by investigating whether the PI3K, MAPK, and NF-κB pathways also contribute to the regulation of ATF3-mediated cellular senescence, SASP expression, periodontal inflammation, and alveolar bone resorption, we can provide a deeper understanding of the molecular mechanisms underlying the role of ATF3 in periodontal disease progression. Further studies on the involvement of these and other signaling molecules and relative pathways in the regulation of ATF3-mediated induction of senescence, inflammation, periodontitis and bone resorption should be performed.
Comments 15: Figure 5A: I suggest to replace the blots where the signal might be over-exposed (STAT3, beta-actin) with blots with a lower signal exposure.
Response 15: According to your comments, we revised it [page 7, Figure 5].
Comments 16: Please make images in fig 5B and 5D bigger in size and with a magnification of 40X as it can be difficult to appreciate the details. Quantification of the number of cells with nuclear signal relative to the total number of cells should be displayed.
Response 16: According to your comments, we revised it [page 7, Figure 5].
Comments 17: Figure 5C: I suggest to replace the lamin B blot as the signal is not comparable in all samples. Please replace all the blots relative to the same experiment. Signals should be quantified and normalized on the loading control.
Please add molecular weights to all blots.
Response 17: According to your comments, we revised it [page 7, Figure 5].
Comments 18: The authors prove that the silencing of ATF3 in PDLCs determines the formation of a lower number of osteoclasts. In line with this, an increase in osteoclastogenesis would be expected in case of overexpression of ATF3, which would be induced by the treatment with LPS and nicotine. Nevertheless, a comparison between BMMs treated with RANKL + M-CSF + CM in control environment (-) and in LPS/nicotine-induced senescence environment (+) does not seem to show this effect.
May the authors explain how ATF3 could behave differently?
Response 18: We sincerely appreciate your insightful comments.
In our study, all osteoclast experiments were performed using conditioned medium (CM) obtained from PDL cells (Figure 3A, B, and C; Figure 4A, B, and C). As shown in Figure 3A–C, both the control and experimental groups were treated with RANKL to induce osteoclast differentiation.
As the reviewer rightly pointed out, it is expected that the LPS+Nicotine CM contains certain cytokines that promote osteoclast differentiation, likely due to the upregulation of ATF3 compared to the control CM. We also acknowledge with regret that, despite this expectation, there was no significant increase in the number of osteoclasts differentiated when compared to the control CM.
However, we observed that in the group treated with LPS+Nicotine CM, multinucleated osteoclasts with fusion of more than five nuclei were more frequently observed. Since we counted osteoclasts based on the presence of three or more nuclei, the total cell number may have appeared similar between groups, but we believe there may have been differences in the size and maturity of the formed osteoclasts. Key genes involved in the fusion of osteoclast precursors, such as DC-STAMP, OC-STAMP, and Atp6v0d2, are known; however, the relationship between ATF3 and these genes has not yet been explored. Moreover, as we did not evaluate these specific genes in our current experiments, we can only hypothesize that ATF3 might be involved in the regulation of mature osteoclast differentiation. We promise to investigate this aspect more deeply in future studies. Once again, we sincerely thank the reviewer for this important suggestion. In future experiments, we will ensure that both the number and size of osteoclasts are carefully measured to better elucidate the differentiation process.
Comments 19: Figure 3B: Could the authors compare statistically the PDLCs grown in control environment (-) and PDLCs grown in LPS/nicotine-induced senescence environment (+) samples?
In addition, the expression of genes involved in osteoclast differentiation, such as Cathepsin K, MMP-9 and NFATc1, is not affected.
May the authors explain these results? May the authors explain how ATF3 can induce osteoclastogenesis if the genes involved in osteoclast differentiation are not downregulated upon its silencing, or upregulated upon ATF3 overexpression?
Response 19: According to your comments, we revised it [page 5, Figure 3].
Upon re-examination of gene expression levels through additional experiments, we observed that mRNA expression of Cathepsin K and MMP-9 was decreased in the experimental group treated with siATF3 and LPS/nicotine-conditioned medium, compared to the groups treated with RANKL + M-CSF + CM (LPS + nicotine or siScr + LPS + nicotine). This pattern was consistent with the results obtained from TRAP staining and osteoclast cell number analysis. Although the expression of the NFATc1 gene was not affected, we aim to further investigate the precise reasons underlying this observation. Specifically, we plan to identify the unknown factors within the conditioned medium (CM) that may suppress osteoclast differentiation genes, and to elucidate how these factors influence osteoclast-related mechanisms. We sincerely appreciate your valuable suggestions and will endeavor to conduct more in-depth studies to address these important questions.
Comments 20: Can the authors replace Con with – or control in the graph in Figure 3C?
Response 20: According to your comments, we revised it [page 5, Figure 3].
Comments 21: Do the authors know the effect of ATF3 silencing on Acp5, c-Fos, tartrate resistant acid phosphate (TRAP), osteoclast-associated receptor (OSCAR) and the d2 isoform of vacuolar ATPase V(0) domain gene expression in BMMs? Can the authors assess osteoclastogenesis by evaluating TRAP activity?
Response 21: We sincerely appreciate your valuable advice and insightful comments.While our current study focused on the senescence and inflammatory phenotype of PDLCs, we fully agree that future studies should address the impact of ATF3 silencing on osteoclastogenesis using BMMs or co-culture systems, and should include detailed analyses of TRAP activity and the expression of osteoclast-related genes. We will endeavor to conduct further in-depth studies to elucidate the precise mechanisms by which ATF3 regulates osteoclastogenesis. We have included this perspective in the revised Discussion section to reflect the importance of these additional investigations [page 9, Discussion, Ref; 25, 35]..
Comments 22: I would like to suggest the authors to investigate deeper the role of ATF3 in PDLCs in LPS/nicotine-induced senescence environment in relation to the differentiation of BMMs into osteoclasts. May the authors provide other findings to support their statement?
Response 22: Agree. We have, accordingly, modified to emphasize this point [Discussion]
We agree that investigating the role of ATF3 in PDLCs within the context of LPS/nicotine-induced senescence and its influence on osteoclast differentiation from BMMs is an important direction.
In our study, we addressed this by performing a bone resorption assay, which demonstrated that conditioned medium (CM) from ATF3-silenced senescent PDLCs significantly reduced both the resorption depth and area on dentin slices (Figure 4A–C). This result suggests that ATF3 in PDLCs contributes to the secretion of factors that promote osteoclastogenesis and bone resorption.
While our current findings support the indirect role of ATF3 in enhancing osteoclast differentiation through secreted factors, we acknowledge that more direct evidence—such as gene expression profiling of osteoclast-specific markers (e.g., Acp5, c-Fos, OSCAR, TRAP, V-ATPase d2) in BMMs treated with CM—would further substantiate this effect.
We have added a statement in the revised Discussion to acknowledge this and highlight the importance of future work.
Comments 23: Do the authors know whether the lower RANKL/OPG ratio in ATF3-silenced PDLCs can be related to a lower abundance of RANKL in in ATF3-silenced PDLCs compared to control cells? Do the authors know whether the effect of PDLCs CM on BMMs differentiation into osteoclasts may be mediated by RANKL? Could they investigate this aspect? RANKL can bind to RANK on the plasmatic membrane of BMMs. RANK activation leads to increased intracellular Ca2+, activation of the c-Fos and NFATs transcription factors that regulate osteoclast-specific gene transcription, their association with ATF3, transcriptional activity and osteoclast differentiation.
Response 23: Thank you for your valuable suggestion. We acknowledge that RANKL is a key mediator of osteoclast differentiation and that the conditioned medium (CM) from PDLCs could influence BMM differentiation via RANKL secretion. However, in the current study, we did not directly assess RANKL levels in the CM derived from senescent PDLCs with or without ATF3 silencing. We have now added a comment in the revised Discussion section to highlight this limitation and propose that future studies should investigate whether ATF3 regulates RANKL expression in PDLCs under LPS/nicotine-induced senescence conditions.
Comments 24: Figure 4: could the authors compare statistically the PDLCs grown in control environment (-) and PDLCs grown in LPS/nicotine-induced senescence environment (+) samples?
Response 24: According to your comments, we revised it [Figure 4].
Comments 25: Figure 4: could the authors compare statistically the PDLCs grown in control environment (-) and PDLCs grown in LPS/nicotine-induced senescence environment (+) samples?
Figure 4B: please explain the quantification indicated as “resorption pit”.
Response 25: According to your comments, we revised it [page 5].
Minor comments:
Comments 1: Line 45: the word periodontitis is repeated twice.
Response 1: Thank you for pointing this out. According to your comments, we revised it [Line 47].
Comments 2: Figure 1: units in the graphs for ATF3 mRNA expression are missing.
Response 2: According to your comments, we revised it [page 1, Figure 1].
Comments 3: Figure 1A legend: legend is not complete.
Response 3: According to your comments, we revised it [Line 82].
Comments 4: Figure 2A-C: units (y axe) are missing.
Response 4: According to your comments, we revised it [page 3, Figure 2].
Comments 5: Lines 106-110: please revise for clarity.
Response 5: According to your regards, we revised it [Line 113-122].
“In this experiment, osteoclast differentiation was induced in mouse bone marrow-derived macrophages (BMMs) using a mixture of osteoclastogenic medium and conditioned medium (CM) from senescent periodontal ligament cells (PDLCs). Senescence was triggered in PDLCs by lipopolysaccharide (LPS) and nicotine, with ATF3 expression silenced in some cells via siRNA before senescence induction. CM from ATF3-silenced senescent PDLCs significantly suppressed osteoclast differentiation in BMMs, while CM from senescent PDLCs without ATF3 knockdown promoted it. These findings suggest that ATF3 plays a crucial role in mediating the pro-osteoclastogenic effects of senescent PDLCs, and its silencing can attenuate the paracrine signals that promote osteoclastogenesis.”
Comments 6: Line 115: MMP9.
Response 6: It's my mistake. I corrected the typo [Line 126].
Comments 7: Figure 3A: the names of the samples are incorrect. The text is misleading.
Response 7: According to your comments, we revised it [page 5, Figure 3].
Comments 8: Figure 3B: replace RNAKL with RANKL.
Response 8: It's my mistake. I corrected the typo [Figure 3B].
Comments 9: Figure 3D: unit (y axe) is missing.
Response 9: According to your comments, we revised it [page 5, Figure 3].
Comments 10: Figure 3: please align text with graphs.
Response 10: According to your comments, we revised it [page 5, Figure 3].
Comments 11: Figure 3 legend at line 133 “Human PDLCs were pretreated with ATF3 for 24 h”: please clearly refer to Scr and ATF3 siRNA treatment.
Response 11: According to your comments, we revised it in figure legends [Line 147].
Comments 12: Figure 4: replace RNAKL with RANKL.
Response 12: It's my mistake. I corrected the typo [Figure 4].
Comments 13: Figure 4a legend at line 146: please replace ATF3 treatment with “Human PDLCs were treated with either Scr or ATF3 siRNA for 24h, …”.
Response 13: According to your comments, we revised it in figure legends [Line 160].
Comments 14: Line 159: please check the sentence “Following the silencing ATF3, …”.
Response 14: According to your comments, we revised it [Line 175].
Comments 15: Lines 164-166: please check the sentence.
Response 15: According to your comments, we checked and revised it. [Line 181].
Comments 16: Line 191: please check the sentence.
Response 16: According to your comments, we revised it [Line 207].
Comments 17: Lines 235-238: the sentence is not clear.
Response 17: According to your comments, we revised it [Line 262].
Comments 18: Line 257: G is missing in the text (in GSE7321).
Response 19: It's my mistake. I corrected the typo [Line 293].
Comments 21: Lines 288-289: please specify the company where the primers have been obtained.
Response 21: According to your comments, we revised it [Line 325].
Comments 22: Line 292: please explain acronym “ICR mice”.
Response 22: The full name for ICR mice is Institute of Cancer Research (ICR) mice. These mice are widely used in research, especially for studies related to cancer, immunology, and pharmacology. The full name has been corrected [Line 330].
Comments 23: Line 300: please check the sentence.
Response 23: According to your comments, we revised it [Line 338].
Comments 24: Line 307: please describe the protocol for preparing nuclear extracts.
Response 24: According to your comments, we revised it [Line 345].
Comments 25: Line 333: please explain all p values; **** stands for p < 0.0001; *** stands for p<0.001; ** stands for p<0.01; * stands for p<0.05.
Response 25: According to your comments, we revised it [Line 376].
Comments 26: Lines 424-427: references 34 and 35 are the same.
Response 26: According to your comments, we delete it [references 35].

Round 2
Reviewer 2 Report
Comments and Suggestions for Authors
The manuscript has been improved
Author Response
We would like to express our sincere gratitude for the time and effort you dedicated to reviewing our revised manuscript. We deeply appreciate your thoughtful and constructive comments during the review process, as well as your positive evaluation of the revised version.

Reviewer 3 Report
Comments and Suggestions for Authors
The presentation of the data and the quality of the manuscript have been improved.
Could the authors include in the manuscript the data on the cell viability or refer to previous findings, if available? I consider it necessary in order to support the results.
I would suggest the authors to consider the possibility of separate each graph into two graphs relative to the two different treatments of the samples (-, +; scr RNA, ATF3 siRNA). I do not consider scientifically relevant the comparisons of the + vs scr RNA samples and + vs siATF3 samples.
I encourage the authors to investigate deeper the role of ATF3 in PDLCs in LPS/nicotine-induced senescence environment in relation to the differentiation of BMMs into osteoclasts. Further in-depth studies to elucidate the precise mechanisms by which ATF3 regulates osteoclastogenesis will provide a stronger support to the present findings and will greatly improve the quality of the manuscript.
Minor comments:
Lines 72-74: please revise the sentence for missing verb.
Lines 76-78 (“Among the concentrations tested (10–30 pmol), the 30 pmol treatment group showed a reduction in ATF3 expression (Figure 1C). Therefore, 20 pmol was selected for subsequent experiments.”): May the authors explain why they have not included the 20pmol treatment in the graph in Figure 1C? How can they prove that its effect would be similar to that of 30 pmol treatment?
Line 84: may the authors report both samples (healthy and patient samples) in the legend in relation to “cells (right) from periodontitis patients”?
Lines 96-97, Figure 2D: may the author quantify the data?
Lines 127-128: please revise the sentence in light of the data shown in figure 3C. May the authors quantify the data shown in figure 3C? May the authors explain why a difference in the expression of cathepsin K and MMP-9 has not been detected upon treatment with LPS + nicotine compared to control sample? Can the authors discuss the role of ATF3 in regulating the expression of these genes?
Line 148: please replace “scranble” with “scramble”.
Lines 169-170: pvalue for *** is missing.
Lines 252-254: Cathepsin K and MMP9 expression is not correctly described.
Lines 264-266 (“Our results demonstrated that Western blot and immunofluorescence analyses revealed that LPS- and nicotine-induced activation of p65 and c-Jun/c-Fos was significantly inhibited in ATF3-deficient PDLCs.”): the results on p65, c-Jun/c-Fos do not seem to strongly support this statement. The quantification of the data in graphs and relative statistical analyses are necessary. For instance, c-Fos abundance do not seem to increase upon LPS + nicotine treatment and to decrease upon ATF3 silencing.
Figure 1: units in the graphs for ATF3 mRNA expression are missing.
Figure 2A-C: units (y axe) are missing.
Figure 2, line 109: p-value for **** is missing.
Figure 3: Could the authors please replace the “Con” words in Figure 3A-C with “–“ or “control”? This would improve the clarity of the data.
Figure 3A legend: scale bar size is missing.
Figure 4A legend: scale bar size is missing.
Figure 5: I suggest to replace the STAT3 blot with one with a lower signal exposure.
Figure 5D: may the authors remove the second scale bar in the last picture at the bottom right of the figure?
Figure 5: may the authors include the graphs relative to the quantifications of data in figure 5? May they perform the statistical analysis? The quantification of the number of cells with nuclear signal relative to the total number of cells is missing for c-Jun.
Author Response
- Summary
Thank you very much for taking the time to review this manuscript. Please find the detailed responses below and the corresponding revisions/corrections highlighted/in track changes in the re-submitted files.
- Questions for General Evaluation - Reviewer’s Evaluation
|
Does the introduction provide sufficient background and include all relevant references? |
Yes |
|
Is the research design appropriate? |
Can be improved |
|
Are the methods adequately described? |
Yes |
|
Are the results clearly presented? |
Can be improved |
|
Are the conclusions supported by the results? |
Must be improved |
- Point-by-point response to Comments and Suggestions for Authors
The presentation of the data and the quality of the manuscript have been improved.
Comments 1: Could the authors include in the manuscript the data on the cell viability or refer to previous findings, if available? I consider it necessary in order to support the results.
Response 1: Thank you for your meaningful comments. Although the data is not shown in the results, we examined the cytotoxicity of PDLCs by treating them with various concentrations of LPS (1 ug/ml) and nicotine ((0, 1, 2.5, 5, 10 mM) for 3 days. Cell viability was measured at 450 nm absorbance using WST (water soluble tetrazolium salt) (EZ-Cytox, BoGenBio Co., Ltd., Korea). It is known that a decrease or arrest of cell proliferation without inducing cell death is a characteristic of cell senescence [2, 16].
Nicotine at low concentrations (0–2.5 mM), when combined with LPS (1 µg/ml), did not cause statistically significant changes in cell viability over 1, 2, or 3 days, as indicated by the "ns" (not significant) labels. However, treatment with 10 mM nicotine consistently resulted in a significant reduction in absorbance (OD 450 nm), suggesting a marked decrease in cell viability. This cytotoxic effect of high-dose nicotine became more pronounced over time. These results indicate that while low-dose nicotine has minimal impact on cell viability in the presence of LPS, high-dose nicotine exerts a significant cytotoxic effect.
Therefore, based on these pilot data, we determined that the combined treatment of LPS (1 ug/ml) + nicotine (5 mM) was appropriate in our study.
[cell viability]
Statistical analysis was conducted by comparing the control group with the nicotine-treated group, respectively. Data are representative of three independent experiments. All data are represented as the mean ±SD. Statistically significant differences were indicated by asterisks in the graphs, with * p < 0.05. ns; No significant difference
Ref [2]: Li, Y.; Tian, X.; Luo, J.; Bao, T.; Wang, S.; Wu, X. Molecular mechanisms of aging and anti-aging strategies. Cell Commun. Signal. 2024, 22, 285. doi: 10.1186/s12964-024-01663-1.
Ref [16]: Zhang, L.; Pitcher, L. E.; Yousefzadeh, M. J.; Niedernhofer, L. J.; Robbins, P. D.; Zhu, Y. Cellular senescence: a key therapeutic target in aging and diseases. J. Clin. Invest. 2022, 132. doi: 10.1172/JCI158450.
Comments 2: I would suggest the authors to consider the possibility of separate each graph into two graphs relative to the two different treatments of the samples (-, +; scr RNA, ATF3 siRNA). I do not consider scientifically relevant the comparisons of the + vs scr RNA samples and + vs siATF3 samples.
Response 2: We appreciate the reviewer’s insightful comment. As suggested, we acknowledge that the Scr RNA sample, although designed as a negative control, did not consistently show behavior similar to the LPS/nicotine-induced senescence (+) group in some experiments (e.g., Figures 2A and 4C). This inconsistency could indeed be due to off-target effects of the siRNA transfection or biological variability.
In response to the reviewer’s recommendation, we revised our analysis strategy. Instead of comparing the ATF3 siRNA group directly to the LPS/nicotine-induced senescence (+) group, we now present the data in two independent comparisons (-, +; scr RNA, ATF3 siRNA)
These changes have been reflected in the revised figures (Figures 2 and 4) and the corresponding figure legends and results section of the manuscript.
In Figure 4, the positions of B and C were modified to match the order of Figure A, and quantitative analysis of the total area of resorption (Figure 4B) was performed again.
Comments 3: I encourage the authors to investigate deeper the role of ATF3 in PDLCs in LPS/nicotine-induced senescence environment in relation to the differentiation of BMMs into osteoclasts. Further in-depth studies to elucidate the precise mechanisms by which ATF3 regulates osteoclastogenesis will provide a stronger support to the present findings and will greatly improve the quality of the manuscript.
Response 3: Thank you for your valuable comments.
In the present study, we aimed to examine the indirect effects on osteoclast differentiation using conditioned media (CM) derived from senescent PDL cells in which ATF3 function was suppressed. Our results indicate that ATF3 regulates SASP factor expression through the STAT3/ERK and NF-κB/AP-1 (c-Jun) signaling pathways in hPDLCs. Notably, suppression of ATF3 resulted in downregulation of RANKL and upregulation of OPG expression at the mRNA level. Based on these observations, we hypothesize that these changes may have led to an altered secretory profile in the CM.
Indeed, osteoclast formation was attenuated in BMMs treated with CM obtained from ATF3-silenced, LPS+nicotine-induced senescent PDL cells. These findings suggest that the composition of osteoclast-regulating factors within the CM was altered. Although we assessed SASP factors and RANKL/OPG expression only at the transcriptional level, the observed suppression of osteoclast differentiation strongly implies that functional components within the CM contributed to this inhibitory effect.
Furthermore, we observed that the expression of osteoclast-associated markers such as Cathepsin K and MMP9 was comparable among the untreated, LPS+nicotine-treated, and siScr CM-treated groups, but was specifically decreased in the siATF3 CM group. This may be attributed to the significant increase in OPG expression observed only in the ATF3-silenced PDLCs, which we believe was sufficient to antagonize RANKL-induced osteoclastogenesis under inflammatory conditions.
We believe that the role of ATF3 in osteoclast differentiation of BMMs in the LPS/nicotine-induced senescence environment of PDLCs is an important topic, and we plan to analyze it more deeply in future studies. The contents were reflected in the manuscripts to supplement the significance of this study [Page 8, Discussion].
Minor comments:
Comments 1: Lines 72-74: please revise the sentence for missing verb.
Response 1: According to your comments, we added 'was observed'[Line 73].
Comments 2: Lines 76-78 (“Among the concentrations tested (10–30 pmol), the 30 pmol treatment group showed a reduction in ATF3 expression (Figure 1C). Therefore, 20 pmol was selected for subsequent experiments.”): May the authors explain why they have not included the 20pmol treatment in the graph in Figure 1C? How can they prove that its effect would be similar to that of 30 pmol treatment?
Response 2: We sincerely apologize for the mistake. The correct amount is 30 pmol, not 20 pmol. We deeply regret the oversight and have now corrected it to 30 pmol in all relevant documents.
Comments 3: Line 84: may the authors report both samples (healthy and patient samples) in the legend in relation to “cells (right) from periodontitis patients”?
Response 3: According to your comments, we revised it [Line 83-85].
“GSE27993 provides ATF3 gene expression data from human periodontal ligament (PDL) tissues of healthy individuals and periodontitis patients (left). GSE7321 focuses on ATF3 gene expression in human PDLCs from healthy and periodontitis-affected individuals (right).”
Comments 4: Lines 96-97, Figure 2D: may the author quantify the data?
Response 4: According to your comments, we revised it [Figure 2D].
Comments 5: Lines 127-128: please revise the sentence in light of the data shown in figure 3C. May the authors quantify the data shown in figure 3C? May the authors explain why a difference in the expression of cathepsin K and MMP-9 has not been detected upon treatment with LPS + nicotine compared to control sample? Can the authors discuss the role of ATF3 in regulating the expression of these genes?
Response 5: Thank you very much for your careful and insightful comment.
We performed quantitative analysis of the Figure 3C PCR bands using ImageJ software, and result statement writing was modified [Figure 3C and Line 128-131].
“The expression levels of Cathepsin K and MMP9 were reduced in the experimental group treated with CM derived from ATF3-silenced PDLCs stimulated with LPS and nicotine, whereas NFATc1 expression showed no significant difference between the groups (Figure 3C).”
In previous studies investigating osteoclast differentiation of BMMs, conditioned medium (CM) was commonly applied in the presence of soluble RANKL at concentrations of either 50-100 ng/mL. In our experiment, we decided to use RANKL at a concentration of 100 ng/mL because we confirmed that osteoclast differentiation was sufficiently induced within 4 days at 100 ng/mL RANKL. To clearly assess the effects of ATF3 inhibition on osteoclastogenesis, we aimed to establish conditions that robustly induce osteoclast differentiation and thus selected the higher concentration of 100 ng/mL RANKL. Therefore, the relatively high concentration of RANKL used in our experiment may have contributed to the absence of significant differences in the expression levels of Cathepsin K and MMP9 among the three groups: the control group directly treated with RANKL and M-CSF, the group treated with LPS+Nicotine CM, and the control CM group.
Jeong et al. reported that ATF3 interacts with c-Fos and NFATc1, participating in calcium signaling pathways involved in osteoclast differentiation and function [29]. These signaling molecules are known to induce the expression of Cathepsin Kand MMP-9. Our study investigated the indirect effects by conditioned medium (CM) derived from ATF3-silenced PDLCs on osteoclast formation in BMMs. Under these experimental conditions, we confirmed that ATF3 regulates the expression of key transcription factors including STAT3, ERK, NF-κB (p65), and c-Jun, which are known to control SASP factors and RANKL expression—critical contributors to osteoclast differentiation and activation. In our senescence model, suppression of ATF3 led to the inhibition of STAT3, ERK, NF-κB, and c-Jun signaling, resulting in decreased secretion of inflammatory SASP cytokines and RANKL, and potentially increased expression of OPG, a decoy receptor that counteracts RANKL activity. These findings suggest that ATF3 knockdown significantly altered the secretory profile of the CM, which may have contributed not only to the suppression of osteoclast differentiation but also to the downregulation of major osteoclastic genes such as Cathepsin Kand MMP-9 in BMMs.
We sincerely thank the reviewer for their thoughtful comments and for providing us with the opportunity to further clarify and strengthen the interpretation of our findings.
Comments 6: Line 148: please replace “scranble” with “scramble”.
Response 6: It's my mistake. I corrected the typo [Line 151, 164].
Comments 7: Lines 169-170: pvalue for *** is missing.
Response 7: According to your comments, we revised it [Line 172].
Comments 8: Lines 252-254: Cathepsin K and MMP9 expression is not correctly described.
Response 8: According to your comments, we revised it [Line 260].
“we observed that the expression of osteoclast-related markers such as cathepsin K and MMP9 was similar in the untreated group, LPS + nicotine treated group, and siScr CM treated group. Although there was no significant change in NFATc1 expression, it was specifically decreased in the siATF3 CM group.”
Comments 9: Lines 264-266 (“Our results demonstrated that Western blot and immunofluorescence analyses revealed that LPS- and nicotine-induced activation of p65 and c-Jun/c-Fos was significantly inhibited in ATF3-deficient PDLCs.”): the results on p65, c-Jun/c-Fos do not seem to strongly support this statement. The quantification of the data in graphs and relative statistical analyses are necessary. For instance, c-Fos abundance do not seem to increase upon LPS + nicotine treatment and to decrease upon ATF3 silencing.
Response 9: Thank you for pointing this out. We agree with this comment. We agree that the current presentation of the data in the Western blot and immunofluorescence images may not clearly demonstrate statistically significant changes in p65, c-Jun and c-Fos activation, particularly regarding c-Fos levels under LPS + nicotine treatment and ATF3 silencing. To address this concern, we performed densitometric analysis of the Western blot bands for p-p65, p-c-Jun and c-Fos [Figure 5C]. Addition, we have revised the text on lines 278–281
“Our results demonstrated that Western blot and immunofluorescence analyses revealed that LPS- and nicotine-induced activation of p65 and c-Jun was significantly attenuated in ATF3-deficient PDLCs, whereas the change in c-Fos levels was not.”
Comments 10: Figure 1: units in the graphs for ATF3 mRNA expression are missing.
Response 10: According to your comments, we revised that the y-axis has been updated to indicate fold change, as the data represent the relative variation compared to the control [Figure 1].
Comments 11: Figure 2A-C: units (y axe) are missing.
Response 11: According to your comments, we revised that the y-axis has been updated to indicate fold change [Figure 2].
Comments 12: Figure 2, line 109: p-value for **** is missing.
Response 12: As the figure did not contain a **** p-value, the corresponding statement has been removed from the figure legend. We deeply regret any confusion caused and appreciate your understanding. [Figure 2].
Comments 13: Figure 3: Could the authors please replace the “Con” words in Figure 3A-C with “–“ or “control”? This would improve the clarity of the data.
Response 13: According to your comments, we revised it [Figure 3].
Comments 14: Figure 3A legend: scale bar size is missing.
Response 14: According to your comments, we revised it [Figure 3 legend].
Comments 15: Figure 4A legend: scale bar size is missing.
Response 15: According to your comments, we revised it [Figure 4 legend].
Comments 16: Figure 5: I suggest to replace the STAT3 blot with one with a lower signal exposure.
Response 16: According to your comments, we revised it [Figure 5A].
Comments 17: Figure 5D: may the authors remove the second scale bar in the last picture at the bottom right of the figure?
Response 17: According to your comments, one of the scale bars has been removed [Figure 5D].
Comments 18: Figure 5: may the authors include the graphs relative to the quantifications of data in figure 5?
May they perform the statistical analysis? The quantification of the number of cells with nuclear signal relative to the total number of cells is missing for c-Jun.
Response 18: Thank you for your meaningful comments. we revised it [Figure 5].
In accordance with the reviewer’s insightful suggestion, the revised content has been added to the Discussion section. We would like to express our sincere gratitude once again for the thoughtful comments and the opportunity to improve our manuscript.
With my best regards and looking forward to hearing from you.
Sincerely yours,
Sang-Im Lee, RDH, PhD, Professor,
Department of Dental Hygiene, College of Health Science, Dankook University
Cheonan 31116, Republic of Korea
Phone number: +82-41-550-1492

Round 3
Reviewer 3 Report
Comments and Suggestions for Authors
The quality of the manuscript has been improved.
I have some minor comments:
Figure 3D: statistical comparison between - and + is missing.
Figure 2D, 3C,5A, 5C: statistical analysis is missing. Are the differences between the scr and siATF3 samples significafive?
Graphs in 5B, 5D: statistical comparison between - and + is missing.
Lines 278-279: western blot in Figure 5 show a similar or decresased value for p65 and c-Jun, respectively, and not a LPS and nicotine-induced activation. Statistical analysis should be performed between the samples - and + in relation to the immunofluorescence experiments in figure 5. Moreover c-Jun nuclear signal does not show a statistically significant decrease upon ATF3 silencing in the immunofluorescence experiment. Please revise the sentence in lines 278-279 according to the results and analysis shown in figure 5.
Author Response
The quality of the manuscript has been improved.
I have some minor comments:
Comments 1: Figure 3D: statistical comparison between - and + is missing.
Response 1: According to the revised it [Figure 3D].
Comments 2: Figure 2D, 3C,5A, 5C: statistical analysis is missing. Are the differences between the scr and siATF3 samples significafive?
Response 2: We acknowledge the omission of statistical analysis in the original submission. Statistical comparisons between scr and siATF3 samples have now been performed and the corresponding statistical annotations have been included directly in Figures 2D, 3C, 5A, and 5C.
Comments 3: Graphs in 5B, 5D: statistical comparison between - and + is missing.
Response 3: According to the revised it [Figure 5].
Comments 4: Lines 278-279: western blot in Figure 5 show a similar or decresased value for p65 and c-Jun, respectively, and not a LPS and nicotine-induced activation. Statistical analysis should be performed between the samples - and + in relation to the immunofluorescence experiments in figure 5. Moreover c-Jun nuclear signal does not show a statistically significant decrease upon ATF3 silencing in the immunofluorescence experiment. Please revise the sentence in lines 278-279 according to the results and analysis shown in figure 5.
Response 4: We agree with the reviewer’s observation, and revised the text in lines 278–279
“ Western blot analysis showed that p65 and c-Jun levels were similar or decreased in the siATF3-treated experimental group, and no activity was observed in response to LPS and nicotine treatment. In addition, immunofluorescence analysis showed that the decrease in nuclear c-Jun signal following ATF3 expression suppression was not statistically significant.”
We hope these revisions address your concerns adequately. Thank you again for your insightful feedback, which helped us improve the clarity and rigor of our manuscript.
With my best regards and looking forward to hearing from you.
